

# Finite-time bounds on the probabilistic violation of the second law of thermodynamics

**Harry J. D. Miller[1] and Martí Perarnau-Llobet[2]**

**1** Department of Physics and Astronomy, The University of Manchester, Manchester M13 9PL, UK
**2** Département de Physique Appliquée, Université de Genève, 1211 Genève, Switzerland

## Abstract

Jarzynski's equality sets a strong bound on the probability of violating the second law of thermodynamics by extracting work beyond the free energy difference. We derive finite-time refinements to this bound for driven systems in contact with a thermal Markovian environment, which can be expressed in terms of the geometric notion of thermodynamic length. We show that finite-time protocols converge to Jarzynski's bound at a rate slower than $1/\sqrt{\tau}$, where $\tau$ is the total time of the work-extraction protocol. Our result highlights a new application of minimal dissipation processes and demonstrates a connection between thermodynamic geometry and the higher order statistical properties of work.

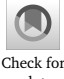

# 1 Introduction

The second law of thermodynamics implies that, on average, the work $W$ extracted from a system in contact with a thermal bath at temperature $T$ is bounded as

$$\langle W \rangle \leq -\Delta F, \tag{1}$$

where $\langle ... \rangle$ stands for an average over realisations of the work-extraction process and $\Delta F$ is the Helmholtz free energy change between an initial and final equilibrium state of the system at temperature $T$. Since $W$ is a stochastic quantity, it is well known that individual events can yield $W > -\Delta F$, i.e., apparent violations of the second law [1–4]. The likelihood of these events is negligible for macroscopic systems but they become relevant at the microscopic level, and this has been a subject of high interest in the last decades [5–20]. When the system is initially at thermal equilibrium, the probability of violating (1) can be inferred from the Jarzynski equality [6], which states that

$$\left\langle e^{\beta W} \right\rangle = e^{-\beta \Delta F}, \tag{2}$$

where $\beta = 1/k_B T$. This equality implies the following bound on the probability $P(W \geq \Lambda)$ of extracting $W \geq \Lambda$ [21],

$$P(W \geq \Lambda) \leq e^{-\beta(\Delta F + \Lambda)}. \tag{3}$$

This means that the likelihood of extracting work above $-\Delta F$ becomes exponentially suppressed as we increase the threshold $\Lambda > -\Delta F$. Recently, Cavina, Mari and Giovannetti [15] developed a work-extraction protocol that saturates the bound (3), which we will refer to as the CMG protocol. It consists of two isothermal processes separated by a quench in the system's energy levels such that the resulting work distribution contains two sharp peaks around the maximum target value $W = \Lambda$ and a minimum $W = W_{\min} \ll -\Delta F$ that represents a worse-case scenario. This structure -two isotherms separated by a quench- is in fact needed to saturate Eq. (3) (see [15] and Sec.5 below). In turn, the presence of the isothermal (and hence reversible) transformations implies that infinite time is required to reach the bound (3) [15,19]. This is in close analogy to the saturation of the standard average law (1), which also requires infinitesimally slow processes. This raises natural questions: What are the fundamental limitations on the probabilistic violation of (1) in finite time? What are the corresponding optimal protocols? How are they related to optimal protocols for maximising $\langle W \rangle$?

The goal of this article is to provide answers to these questions. Focusing on driven systems in contact with a (Markovian) thermal bath, and using techniques from finite time stochastic and quantum thermodynamics [22–24], our key result is to derive a finite-time correction to Eq. (3) that behaves as:

$$P(W \geq \Lambda) \leq e^{-\beta(\Delta F + \Lambda)} - \frac{C_\Lambda}{\tau^\alpha} + \mathcal{O}\left(\tau^{-2\alpha}\right), \qquad 0 < \alpha < 1/2, \tag{4}$$

where $C_\Lambda \geq 0$ and $\tau$ is the total time of the work-extraction process (see Fig. 3). The first implication of (4) is that the convergence to the upper bound (3) in the infinitely slow limit scales no faster than $\mathcal{O}(\tau^{-1/2})$. This demonstrates that even minor deviations from a perfect isotherm can have a noticeable effect on the maximum possible cumulative distribution $P(W \geq \Lambda)$ for any chosen $\Lambda > -\Delta F$. For the finite-time correction, we derive an expression that depends only on the boundary conditions of the protocol (in analogy with $\Delta F$). We also show that the optimal process saturating (4) is one that minimises the average entropy production along the finite-time isotherms, which also ensures the majority of the total work fluctuations are provided by the energetic quench that separates the two isotherms in the protocol. Paths of

minimal entropy production can be found using geometric methods [23, 25, 26], with the optimal protocol corresponding to a geodesic path within the manifold of control parameters. This implies that $C_\Lambda$ in (4) can be expressed in terms of the so-called thermodynamic length [27, 28] between the different boundary points in the protocol.

Our results are directly relevant for a recent implementation of the CMG protocol in a single electron transistor [18], where it was demonstrated that one can extract significant amounts of work beyond the free energy decrease with a probability greater than 1/2. As expected, because the experiment was realised in finite time, the idealised sharp peaks of the CMG protocol were broadened leading to a smaller $P(W \geq \Lambda)$ than theoretically possible. Our results provide means of improving the protocol realised in [18] in order to substantially increase $P(W \geq \Lambda)$ given the same amount of time and level of control. Beyond this interesting application, the optimal processes derived here yield new insights for the control of microscopic machines such as biomolecular motors and single enzymes [2, 29], where one may need to drive the system above some barrier or activation energy that exceeds the available free energy change.

The structure of the paper is as follows: in Section 2 we recall the CMG protocol that can saturate the bound (3) using perfect isothermal steps, in Section 3 we formulate our finite-time version of the protocol for a classical bit, in Section 4 we show how to optimise $P(W \geq \Lambda)$ over all protocols and present a finite-time correction to (3) in terms of the thermodynamic length, and in Section 5 we show how to extend our analysis to higher dimensional systems with non-trivial relaxation dynamics.

## 2 Optimal protocol in infinite time

Here we first give an overview of the CMG protocol developed in [15] that is able to maximise the work extraction likelihood according to (3), given some fixed initial and final equilibrium configuration. For simplicity, and to connect our results directly to the experiment [18], we will first consider transforming a classical bit or two-level quantum dot. Our results will be generalised to systems with more degrees of freedom in Sec. 5. Suppose our classical bit has two distinct energy levels, $\{0, E\}$ with $E > 0$, and when in equilibrium at inverse temperature $\beta$ the probability of observing the system in the zero-energy ground state is

$$p(E) := \frac{1}{1 + e^{-\beta E}}. \tag{5}$$

The excited energy level $E$ represents the free parameter that can be controlled externally in order to extract work. The free energy of the bit is given in terms of $E$ as

$$F(E) = -\frac{1}{\beta} \ln \left( 1 + e^{-\beta E} \right). \tag{6}$$

The CMG protocol consists of three steps (see Fig. 1):

(A) The system begins with spacing $E_i$ and undergoes an isothermal process at inverse temperature $\beta$ by changing the Hamiltonian to a new value $E_a$.

(B) The system Hamiltonian is then quenched rapidly from $E_a$ up to a greater value $E_b > E_a$ with no dissipation arising from the environment.

(C) Another isothermal process at the same temperature is performed from energy $E_b$ to $E_f$.

During the isotherm in Step (A), the extracted work is given deterministically by the free energy decrease $W_A = F(E_i) - F(E_a)$. In Step (B), we either extract no additional work $W_B = 0$ with probability $p(E_a)$ as defined in (5) or an amount $W_B = E_a - E_b$ with probability $1 - p(E_a)$ if

there is an excitation of the bit. In the final isothermal Step ($C$), we extract work equal to the free energy decrease $W_C = F(E_b) - F(E_f)$ deterministically. The total work extracted is just the sum of these three steps,

$$W := W_A + W_B + W_C \,, \tag{7}$$

and the resulting work distribution takes the form

$$P(W) = p(E_a)\delta\big[W - W_{\max}\big] + \big(1 - p(E_a)\big)\delta\big[W - W_{\min}\big] \,, \tag{8}$$

where

$$W_{\max} = -\Delta F + \Delta F_{ab} \,, \tag{9}$$

$$W_{\min} = W_{\max} - \Delta E_{ab} \,. \tag{10}$$

Here we define $\Delta F = F(E_f) - F(E_i)$ as the total free energy change, $\Delta F_{ab} = F(E_b) - F(E_a)$ the intermediate free energy change across Step $B$ and $\Delta E_{ab} = E_b - E_a$ the work *done* during Step $B$. It is straightforward to see from $E_b > E_a$ that the two peaks in the work distribution are situated either side of the total free energy decrease, namely $W_{\max} > -\Delta F$ and $W_{\min} < -\Delta F$. It is then clear that

$$P(W \geq W_{\max}) = \int_{W_{\max}}^{\infty} dW \; P(W) = p(E_a) = \frac{e^{-\beta \Delta F} - e^{\beta W_{\min}}}{e^{\beta W_{\max}} - e^{\beta W_{\min}}} \,, \tag{11}$$

where the final equality follows from inverting the pair of equations (9) and (10) to solve for $E_a$. Note that in a situation where we have an additional constraint on the work distribution $P(W < W_{\min}) = 0$, the optimal bound (3) is corrected to [15]

$$P(W \geq \Lambda) \leq \frac{e^{-\beta \Delta F} - e^{\beta W_{\min}}}{e^{\beta \Lambda} - e^{\beta W_{\min}}} \,. \tag{12}$$

With the CMG protocol we can then get exponentially close to the upper bound (3) for any threshold $\Lambda$ by choosing $E_a$ and $E_b$ such that

$$W_{\max} = \Lambda \,, \quad \text{and} \quad W_{\min} \to -\infty \,. \tag{13}$$

Importantly, given a two level system, the described protocol is the only one saturating the bound (3) [15], see Sec. 5 for generalisations to $d$-level systems.

    The physical limitation of the CMG protocol rests on the fact that Steps ($A$) and ($C$) require perfect isothermal transformations. Isothermal processes in principle require an infinite amount of time as the system must remain in equilibrium with respect to the environment at all times. However, in any realistic implementation these steps will occur with a finite time duration, in which case the system will deviate from perfect equilibrium. This introduces additional dissipation and fluctuations, so that we can no longer expect the extracted work to equal $W_A = F(E_i) - F(E_a)$ and $W_C = F(E_b) - F(E_f)$ with zero stochastic fluctuations during the isotherms [19, 30–33]. This finite-time behaviour was observed in the experiment in [18] where the two peaks in (8) were replaced by a pair of normal distributions of finite width as illustrated in Fig. 1; see also the recent experiment [34] and the theoretical work [19] for similar effects. These additional fluctuations prevent one from obtaining the optimal bound (12). In the next section, we show how to model this behaviour for regimes where the system remains close to isothermal.

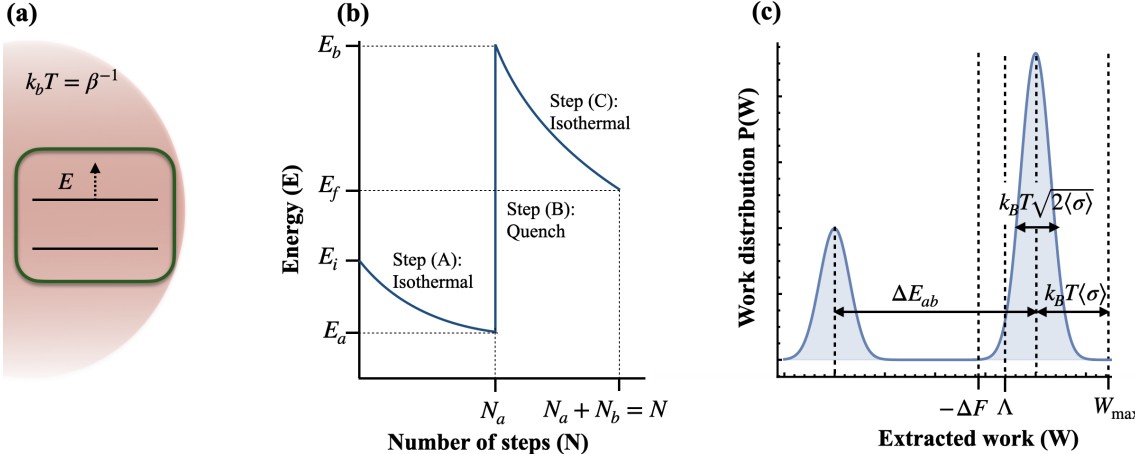

Figure 1: **(a)** Set-up considered in this work: Work extraction through a driven two-level system in contact with a thermal bath at inverse temperature $\beta$ (see also Sec. 5 for extensions). The second law of thermodynamics implies that $\langle W \rangle \leq -\Delta F$, but individual events can satisfy $W > -\Delta F$. Our goal is to maximise the likelihood of such events, given by $P(W \geq \Lambda)$ with $\Lambda \geq -\Delta F$. **(b)** Sketch of the CMG work extraction protocol, consisting of two isothermal/reversible processes (from $E_i$ to $E_a$, and from $E_b$ to $E_f$) separated by a quench from $E_a$ to $E_b$ [15]. This protocol can maximise probabilistic work extraction, $P(W \geq \Lambda)$, and saturate the fundamental bound (3). However, it requires infinite time to realise perfect isotherms ($N \to \infty$). **(c)** A sketch of the work probability distribution $P(W)$ obtained when performing the CMG protocol in finite time (finite $N$). It consists of two Gaussian distributions with their first and second moments related via the work fluctuation dissipation relation [33]. In the limit $N \to \infty$, the (average) entropy production (26) vanishes $\langle \sigma \rangle = 0$ so that the Gaussian distributions are replaced by Dirac deltas; the protocol maximising $P(W \geq \Lambda)$ then requires choosing $E_b \to \infty$ and $E_a$ such that $\Lambda = W_{\max}$ (13). Instead, for finite time $\langle \sigma \rangle > 0$, it is convenient to take finite $E_b$ and $E_a$ such that $W_{\max} > \Lambda$ in order to maximise $P(W \geq \Lambda)$.

## 3 Probabilistic work extraction in finite time

We now model the steps $(A)$ and $(C)$ in *finite time* by assuming that the system is driven by a series of discrete quenches in the energy $E$ followed by relaxations with respect to the environment, following a well-known discrete approach to finite-time thermodynamics [28, 33, 35, 36]. A similar approach has been recently used in Ref. [19] to characterise $P(W \geq \Lambda)$ as well as the form of the work distribution. We will generalise our results to driven Markovian systems in Sec. 5.

Let us first decompose Step $(A)$ into a series of $N_A - 1$ quenches in the energy gap $E$, where we label the $n$'th energy value by $E_n^{(A)}$ with boundary conditions $E_1^{(A)} = E_i$ and $E_{N_A}^{(A)} = E_a$. The work extracted along the $n$'th quench is given by the energy decrease labelled $W_n^{(A)}$, and the total extracted work along the whole of Step $(A)$ is then just the sum $W_A = \sum_{n=1}^{N_A-1} W_n^{(A)}$. For our two level system, there are two possible outcomes at each quench given by $W_n^{(A)} = 0$ with probability $p(E_n^{(A)})$ and $W_n^{(A)} = E_n^{(A)} - E_{n+1}^{(A)}$ with probability $1 - p(E_n^{(A)})$. Each quench is followed by a relaxation with the environment at zero work cost and each work increment $W_n^{(A)}$ is independent of the previous step, meaning that the likelihood of extracting a given

amount $W_A$ during Step $(A)$ is given by

$$P(W_A) = \prod_{n=1}^{N_A-1} \left( \delta[W_n^{(A)}]p(E_n^{(A)}) + \delta[W_n^{(A)} + E_{n+1}^{(A)} - E_n^{(A)}](1 - p(E_n^{(A)})) \right). \qquad (14)$$

In a similar manner, we can model Step $(C)$ as another series of $N_C - 1$ quenches and relaxations, with the energy gaps passing through a trajectory of values $E_n^{(C)}$ with boundary conditions $E_1^{(C)} = E_b$ and $E_{N_C}^{(C)} = E_f$. The distribution of work $W_C$ during this final step is then, in analogy with (14),

$$P(W_C) = \prod_{n=1}^{N_C-1} \left( \delta[W_n^{(C)}]p(E_n^{(C)}) + \delta[W_n^{(C)} + E_{n+1}^{(C)} - E_n^{(C)}](1 - p(E_n^{(C)})) \right). \qquad (15)$$

Step $(B)$ is also independent of $(A)$ and $(C)$, which means the total work distribution along all three stages with outcome $W = W_A + W_B + W_C$ is given by

$$P(W) = \delta[W - W_A - W_C]P(W_A)P(W_C)p(E_a) \qquad (16)$$
$$+ \delta[W - W_A - W_C + \Delta E_{ab}]P(W_A)P(W_C)(1 - p(E_a)), \qquad (17)$$

where we have used the fact that $W_A$, $W_B$ and $W_C$ are independent random variables, and recall the definition $\Delta E_{ab} = E_b - E_a$.

The degree to which we can get close to the upper bound (3) depends on how close we can approximate the desired isotherms in Step $(A)$ and $(C)$. A perfect isotherm is achieved in the infinite step limit $N_A = N_C = \infty$. In order to investigate the more realistic situation where these steps are finite, while keeping the problem tractable, we turn our attention to a regime where the number of steps are large:

$$N_A^2 \gg 1, \qquad N_C^2 \gg 1, \qquad (18)$$

which means we treat terms of order $\mathcal{O}(1/N_A^2)$ and $\mathcal{O}(1/N_C^2)$ as negligible in subsequent calculations. To this order of approximation, we can replace the series of quenches $\{E_n^{(A)};\ n = 1, 2, ...N_A\}$ and $\{E_n^{(C)};\ n = 1, 2, ...N_C\}$ along Steps $(A)$ and $(C)$ by a pair of smooth functions $E^{(A)}(t)$ and $E^{(C)}(t)$ for dimensionless parameter $t \in [0, 1]$ with fixed boundary conditions (see e.g. [33])

$$\{E^{(A)}(0) = E_i,\ E^{(A)}(1) = E_a\}, \qquad (19)$$
$$\{E^{(C)}(0) = E_b,\ E^{(C)}(1) = E_f\}. \qquad (20)$$

In terms of the full statistics, it is known that the work distribution behaves approximately Gaussian where the average excess work is proportional to half the variance divided by $k_B T$ [30, 33]. That is, for large $N^2 \gg 1$ we can approximate the distribution at Step $(A)$ and $(C)$ by

$$P(W_X) \simeq \mathcal{N}\left(\langle W_X \rangle, -2k_B T\left(\langle W_X \rangle + \Delta F_X\right)\right); \qquad X = \{A, C\}, \qquad (21)$$

where $\mathcal{N}(\langle x \rangle, \Delta x^2)$ denotes a normal distribution with mean $\langle x \rangle$ and variance $\Delta x^2 = \langle x^2 \rangle - \langle x \rangle^2$. In the above we have labelled the free energy changes by $\Delta F_A = F_a - F_i$ and $\Delta F_C = F_f - F_b$, and used the work fluctuation-dissipation relations,

$$\langle W_X \rangle + \Delta F_X = -\frac{1}{2}\beta \Delta W_X^2; \qquad X = \{A, C\}, \qquad (22)$$

which holds up to order $\mathcal{O}(1/N_X^2)$. Comparing this with (16) we see that the full work distribution will be a convex sum of two independent Gaussian distributions centered on $\langle W_A \rangle + \langle W_C \rangle$ and $\langle W_A \rangle + \langle W_C \rangle - \Delta E_{ab}$ respectively. We can now directly compute the likelihood of extracting work above some threshold $\Lambda$, which is given by

$$
\begin{aligned}
P(W \geq \Lambda) &= \int_\Lambda^\infty dW \; P(W) \\
&= \frac{1}{2} p(E_a) \, \text{erfc}\left( \frac{\Lambda - \langle W_A \rangle - \langle W_C \rangle}{2\sqrt{k_B T(W_{\max} - \langle W_A \rangle - \langle W_C \rangle)}} \right) \\
&\quad + \frac{1}{2}\big(1 - p(E_a)\big) \, \text{erfc}\left( \frac{\Lambda + \Delta E_{ab} - \langle W_A \rangle - \langle W_C \rangle}{2\sqrt{k_B T(W_{\max} - \langle W_A \rangle - \langle W_C \rangle)}} \right) \qquad (23) \\
&\simeq \frac{1}{2} p(E_a) \, \text{erfc}\left( \frac{\Lambda - \langle W_A \rangle - \langle W_C \rangle}{2\sqrt{k_B T(W_{\max} - \langle W_A \rangle - \langle W_C \rangle)}} \right), \qquad (24)
\end{aligned}
$$

where

$$
\text{erfc}(x) := \frac{1}{\sqrt{2\pi}} \int_x^\infty dy \; e^{-y^2/2}, \qquad (25)
$$

is the complementary error function and $W_{\max}$ is given in Eq. (9). In the last line of (23) we neglect the second error function as it is exponentially small term with respect to $\beta \Delta E_{ab}$. It is useful to rewrite this in terms of the average entropy production (of steps $A$ and $C$), which is given by

$$
\langle \sigma \rangle = \beta\big(W_{\max} - \langle W_A \rangle - \langle W_C \rangle\big). \qquad (26)
$$

Comparing with (23) gives us

$$
P(W \geq \Lambda) = \frac{1}{2} p(E_a) \, \text{erfc}\left( \frac{\beta(\Lambda - W_{\max}) + \langle \sigma \rangle}{2\sqrt{\langle \sigma \rangle}} \right). \qquad (27)
$$

Given that we now have an expression for the cumulative work distribution, we can optimise it given a fixed free energy change. Let us fix the boundary points according to (19) and (20). Our first main result is the following observation:

*'For large N, the process that maximises the likelihood of extracting work $W \geq \Lambda > -\Delta F$ is one that minimises the total average dissipation $\langle \sigma \rangle$ along Steps (A) and (C).'*

To prove the above statement, let us consider the function

$$
f(x) = \text{erfc}\left( \frac{X + x}{2\sqrt{x}} \right), \qquad (28)
$$

where $X \in \mathbb{R}$ is a fixed constant and $x \geq 0$. Its derivative is given by

$$
f'(x) = -\frac{1}{\sqrt{8\pi}} \left( \frac{x - X}{x^{3/2}} \right) \exp\left( \frac{(X + x)^2}{x} \right), \qquad (29)
$$

which is negative for $X < 0$. If we compare this with (27), we note that $\Lambda \leq W_{\max}$ (see Figure 1), in which case the function $P(W \geq \Lambda)$ is monotonically decreasing in $\langle \sigma \rangle$ as desired.

Fortunately this means that in order to find a finite time correction to the Jarzynski bound (3), we only need to consider deriving a finite time correction to the usual second law bound $\langle \sigma \rangle \geq 0$. We will use some existing geometric techniques [23, 25, 26] to do this in the next section.

# 4 Geometric optimisation of the protocol

To find a process that minimises the average dissipation, it is instructive to introduce some tools from information geometry. Let the vector $\vec{p} = \{p_1, p_2, \ldots p_n\}$ denote a normalised probability distribution with $\sqrt{\vec{p}} \cdot \sqrt{\vec{p}} = 1$ and a set of $d$ outcomes. We can define a line element $ds$ on the manifold of normalised distributions according to

$$ds^2 = \sum_{x=1}^{d} \frac{(dp_x)^2}{p_x}. \tag{30}$$

If we let this distribution depend on some scalar parameter $E$ through $\vec{p} = \vec{p}(E)$ and consider a path $\gamma : t \mapsto E(t)$ for $t \in [0,1]$, the length between points $E(0) = E_i$ and $E(1) = E_f$ is given by

$$l_\gamma = \int_\gamma ds = \int_0^1 dt \, \dot{E}(t) \left( \frac{ds}{dE} \right). \tag{31}$$

It is well known that the shortest curve, or geodesic, connecting the endpoints is given by the Bhattacharyya angle between the initial and final distribution $\vec{p}_i = \vec{p}(E_i)$ and $\vec{p}_f = \vec{p}(E_f)$ respectively [23, 37]. This is given by

$$\inf_\gamma l_\gamma = 2 \arccos \left( \sqrt{\vec{p}_i} \cdot \sqrt{\vec{p}_f} \right). \tag{32}$$

These geometric quantities connect to our setup when one Taylor expands the average dissipation (26) and neglects terms of order $\mathcal{O}(1/N^2)$ [28, 35, 38, 39]. In this case one finds

$$\langle \sigma \rangle \simeq \frac{1}{2N_A} \int_0^1 dt \, \left( \dot{E}^{(A)} \right)^2 \left( \frac{ds^{(A)}}{dE^{(A)}} \right)^2 + \frac{1}{2N_C} \int_0^1 dt \, \left( \dot{E}^{(C)} \right)^2 \left( \frac{ds^{(C)}}{dE^{(C)}} \right)^2, \tag{33}$$

where $ds^{(A)}$ is the line element (30) associated to the binary distribution $\vec{p}(E^{(A)}) = \{p(E^{(A)}), 1 - p(E^{(A)})\}$ as defined by (5) while $ds^{(C)}$ relates to $\vec{p}(E^{(C)}) = \{p(E^{(C)}), 1 - p(E^{(C)})\}$.

Consider now a pair of paths $\gamma_A : t \mapsto E^{(A)}(t)$ and $\gamma_C : t \mapsto E^{(C)}(t)$ describing the chosen protocols along Step $(A)$ and $(C)$ respectively. It follows from the Cauchy-Schwarz inequality that we can tightly lower bound (33) by the sum of squared lengths (32) according to

$$\langle \sigma \rangle \geq \frac{\mathcal{L}_A^2}{2N_A} + \frac{\mathcal{L}_C^2}{2N_C}, \tag{34}$$

where

$$\mathcal{L}_A = 2 \arccos \left( \sqrt{\vec{p}(E_i)} \cdot \sqrt{\vec{p}(E_a)} \right), \tag{35}$$

$$\mathcal{L}_C = 2 \arccos \left( \sqrt{\vec{p}(E_b)} \cdot \sqrt{\vec{p}(E_f)} \right), \tag{36}$$

are the geodesic lengths along $(A)$ and $(C)$. We can saturate this inequality by keeping the integrands in (33) constant [25], which for our two-level system is achieved by choosing a protocol such that

$$\beta \dot{E}^{(X)}(t) \propto \sqrt{\cosh \left[ \beta E^{(X)}(t) \right] + 1}, \quad X = \{A, C\}. \tag{37}$$

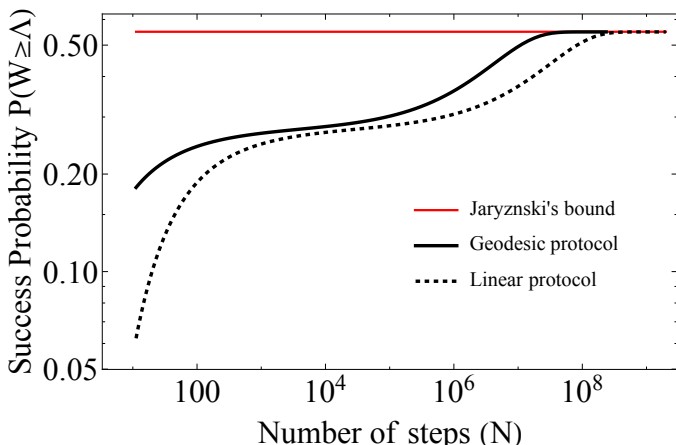

Figure 2: Comparison of $P(W \geq \Lambda)$ for a geodesic path (see Eq. (37)) and following a linear drive of $E(t)$. Parameters: $\beta = 2$, $E_i = 6$, $E_f = 0.1$, $\Lambda = -2\Delta F$, $E_a = 0.987 E_a^\infty$, $E_b \gg 1$.

As a further parameter to consider, we minimise the RHS of (34) with respect to the number steps along (A) and (C) subject to a constraint on the total number of steps $N = N_A + N_C$. The minimum is found by setting

$$N_A = N \frac{\mathcal{L}_A}{\mathcal{L}_A + \mathcal{L}_C}, \tag{38}$$

$$N_C = N \frac{\mathcal{L}_C}{\mathcal{L}_A + \mathcal{L}_C}, \tag{39}$$

and thus the minimal entropy production is given by

$$\langle \sigma^* \rangle := \min_{\gamma_A, \gamma_C} \langle \sigma \rangle = \frac{1}{2N} \left( \mathcal{L}_A + \mathcal{L}_C \right)^2, \qquad N = N_A + N_C. \tag{40}$$

As our final step, we use our observation that the path of minimal dissipation will maximise the chance of extracting work in excess of the free energy and combine (27) with (34) to get

$$P^*(W \geq \Lambda) := \max_{\gamma_A, \gamma_C} P(W \geq \Lambda) = \frac{1}{2} p(E_a) \, \mathrm{erfc}\left( \frac{\beta(\Lambda - W_{\max}) + \langle \sigma^* \rangle}{2\sqrt{\langle \sigma^* \rangle}} \right). \tag{41}$$

This gives us a path independent bound on the optimal probability for work extraction in terms of the four boundary points, $E_i \mapsto E_a \mapsto E_b \mapsto E_f$. Importantly we know that this bound is tight and can be saturated by following the relevant geodesic paths during Step (A) and (C) and choosing the number of steps accordingly. In order to illustrate the relevance of following a geodesic path, in Fig. 2 we compare $P^*(W \geq \Lambda)$ with the $P(W \geq \Lambda)$ obtained via a linear drive $\dot{E}^{(X)}(t) \propto$ constant given some $\{E_a, E_b, N\}$.

As can be seen in Fig. 1, the choice of the intermediate levels $E_a$ and $E_b$ fixes the position of the peak that sits below the free energy change. We can therefore view these boundary points as setting the threshold at which we are willing to tolerate a sub-optimal outcome. For example, while we can ensure that the majority of trajectories will extract some work above $-\Delta F$, this can come at the expenses of having a small chance of consuming a larger amount of work instead. Increasing the chance of optimal work extraction means we have to increase the shift $\Delta E_{ab}$, but then we pay a higher price for the sub-optimal outcomes due to this trade-off. If one is not concerned about the magnitude of the peak below the free energy decrease, then the bound (41) can be further optimised over the intermediate levels $E_a, E_b$:

$$P_{\max}(W \geq \Lambda) := \max_{E_a, E_b} P^*(W \geq \Lambda). \tag{42}$$

In the infinite-time limit $N \to \infty$, we have $P_{\max}(W \geq \Lambda) = e^{-\beta(\Delta F + \Lambda)}$ with the corresponding optimal $E_a, E_b$ given by $E_b \to \infty$ and $E_a = E_a^\infty$ with [15]

$$E_a^\infty \equiv -\frac{1}{\beta} \ln\left(e^{\beta(\Delta F + \Lambda)} - 1\right). \tag{43}$$

For finite $N$, these optimal points can be found numerically, and in general depend on the boundary conditions and $N$. The result of this numerical optimisation is shown in Fig. 3, where we plot $P_{\max}(W \geq \Lambda)$. We stress that $P_{\max}(W \geq \Lambda)$ is optimised over all protocols, and hence can be understood as a finite-$N$ correction of the ultimate bound (3).

We can gain some insight into how $P_{\max}(W \geq \Lambda)$ converges to the Jarzynski bound (3) as a function of the step size $N$ as follows. First recall that in order to approach the infinite time protocol saturating (3) we need $E_b \to \infty$ and $E_a = E_a^\infty$, leading to

$$P(W \geq \Lambda) \to p(E_a^\infty). \tag{44}$$

This means that in the large $N$ limit we require the error function in (41) to approach unity, which implies

$$\lim_{N \to \infty} \left( \frac{\sqrt{\langle \sigma \rangle}}{\beta(\Lambda - W_{\max}) + \langle \sigma \rangle} \right) = 0, \tag{45}$$

or equivalently

$$\lim_{N \to \infty} \left( \frac{1}{\sqrt{N}\beta(\Lambda - W_{\max}) + \langle \sigma \rangle} \right) = 0. \tag{46}$$

In addition to this, to saturate (3) we also need

$$\lim_{N \to \infty} \beta(\Lambda - W_{\max}) + \langle \sigma \rangle = 0. \tag{47}$$

We therefore take an *ansatz*

$$\beta(\Lambda - W_{\max}) + \langle \sigma \rangle = -\frac{\zeta}{N^\alpha}, \tag{48}$$

where $0 < \alpha < 1/2$ and $\zeta > 0$ (positivity can be seen from Fig. 1). Rearranging the LHS in terms of $p(E_a)$, taking $E_b \gg E_a$ and expanding for large $N$ we find

$$p(E_a) = e^{-\beta(\Delta F + \Lambda)} e^{-\xi/N^\alpha} + \mathcal{O}(1/N). \tag{49}$$

Note that if we expand the complimentary error function in (41), we get an exponentially small contribution since

$$\mathrm{erfc}\left( \frac{\beta(\Lambda - W_{\max}) + \langle \sigma^* \rangle}{2\sqrt{\langle \sigma^* \rangle}} \right) \sim \mathcal{O}\left( e^{-N^{1-2\alpha}\zeta^2} \right). \tag{50}$$

Plugging (49) and (50) into (41), we find the leading order corrections to (3) must take the form

$$P_{\max}(W \geq \Lambda) := e^{-\beta(\Delta F + \Lambda)} \left( 1 - \frac{\xi}{N^\alpha} + \mathcal{O}(1/N^{2\alpha}) \right). \tag{51}$$

Hence, while we do not have an analytic expression for $\xi$, we can still conclude that the optimal finite-time protocol will converge to the infinite-time limit (3) as we increase the number of steps at a rate that is always slower than $1/\sqrt{N}$. This confirms the implicit structure of our bound in (4) presented at the start, where in this case we quantify the duration of the process by the number of steps $N$. This slow convergence demonstrates that finite time constraints can lead to a significant correction to the Jarzynski bound (3). In particular, for the specific parameters of Fig. 3 we numerically find that $\alpha \approx 0.44$ and $\zeta \approx 2.45$ is a good approximation for $N \gg 1$. Of course, our analysis here also sets a limit on the speed of convergence of our analytic bound (41) with fixed boundary points as well as any sub-optimal protocol.

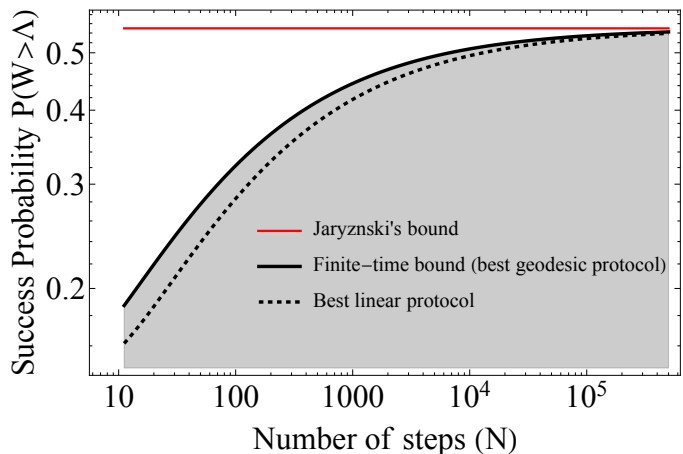

Figure 3: In this figure we illustrate the main result of this article, a finite-time bound on $P(W \geq \Lambda)$, which tends to Jarzynski's bound in Eq. 3 in the asymptotic limit. More precisely, we plot $P_{\max}(W \geq \Lambda)$ given in (42), which provides a bound on all possible protocols, as a function of the number of steps $N$. For comparison, we also show the best linear protocol, obtained by optimising over $E_a$ and $E_b$ all protocols with a linear driving of $E(t)$. Parameters: $\beta = 2.$, $E_i = 6.$, $E_f = 0.1$, $\Lambda = -2\Delta F$.

## 5 Generalisations for multiple energy levels and relaxation timescales

While we have thus far modelled the working substance as a single classical bit, here we generalise our approach to $d$-level systems (qudits) interacting with a Markovian environment. In this case, we replace the discrete step processes by continuously driven open systems for a total time $\tau$ (note that the former can be seen as a particular case of the latter in the slow driving limit [33]). Let us model the system with a Hamiltonian of the form

$$H[\vec{E}(t)] := \sum_{n=1}^{d} E_n(t)|n\rangle\langle n| . \tag{52}$$

Here we assume that there is full control over a finite set of $d$ energy levels $\vec{E}(t) = \{E_1(t), E_2(t), ..., E_d(t)\}$, while each energy eigenstate $|n\rangle$ is kept fixed (in the slow driving limit, rotating the eigenstates would only increase dissipation and work fluctuations [23,33]). For simplicity we assume that there are no degeneracies in the energy levels. The system can be brought in weak contact with a thermal environment such that the corresponding dynamics of its density matrix $\rho(t)$ obeys a time-dependent Markovian master equation, given by

$$\dot{\rho}(t) = \mathscr{L}_{\vec{E}(t)}[\rho(t)] . \tag{53}$$

Here $\mathscr{L}_{\vec{E}(t)}[.]$ is a time-dependent generator parameterised by the set of energy levels, and has an instantaneous thermal fixed point:

$$\mathscr{L}_{\vec{E}(t)}[\pi(\vec{E}(t))] = 0; \qquad \pi(\vec{E}(t)) = \frac{e^{-\beta H[\vec{E}(t)]}}{\text{Tr}(e^{-\beta H[\vec{E}(t)]})} . \tag{54}$$

Furthermore, the free energy of the equilibrium state is given by

$$F(\vec{E}) := -k_B T \ln \text{Tr}(e^{-\beta H[\vec{E}]}) . \tag{55}$$

Let us first discuss the family of protocols that can saturate Eq. (12) (and hence (3)) in the asymptotic limit $\tau \to \infty$. This bound can only be saturated by a two-peak distribution of the form (8) [15]. In turn, this can only be achieved by a protocol of the form in Fig. 1, i.e., two isothermal processes separated by a quench. The isothermal processes ensure the lack of fluctuations in each peak of the work distribution [15,33], whereas the quench is needed to obtain the two peaks. In analogy with Sec. 2, we consider Steps ($A$) and ($C$) that involve changing the energy levels according to two smooth curves $\gamma_A : t \mapsto \vec{E}^{(A)}(t)$ and $\gamma_C : t \mapsto \vec{E}^{(C)}(t)$ each with duration $t \in [0, \tau_A]$ and $t \in [0, \tau_C]$ respectively, and with $\tau = \tau_A + \tau_C$. Step (B) connects $\vec{E}^{(A)}(\tau_A)$ with $\vec{E}^{(C)}(0)$ by an energy quench (where the state is assumed not to evolve). As we did with (19), the boundary conditions for the three steps are taken as

$$\{\vec{E}^{(A)}(0) = \vec{E}_i , \; \vec{E}^{(A)}(\tau_A) = \vec{E}_a\}, \tag{56}$$

$$\{\vec{E}^{(C)}(0) = \vec{E}_b , \; \vec{E}^{(C)}(\tau_C) = \vec{E}_f\}. \tag{57}$$

$\vec{E}_i$ and $\vec{E}_f$ are given by the particular work-extraction process of interest (they define $\Delta F$), whereas $\vec{E}_a$ and $\vec{E}_b$ can be chosen to maximise probabilistic work extraction. In order to replicate the two-peak work distribution (8), we need the following constraint

$$\vec{E}_b = \vec{E}_a + \lambda \vec{\delta}, \quad \text{with} \quad \delta_j \in \{0, 1\} \; \forall j, \tag{58}$$

where the vector $\vec{\delta}$ containing only 0 and 1's can be chosen at will (for qudit systems, the optimal protocol is not unique), and the constant $\lambda$ plays the analogous role of $\Delta E_{ab}$ for the qubit system (see Eqs. (10) and 9).

Having characterised the family of optimal protocols for qudit systems, we now derive finite-time corrections in analogy with Secs. 3 and 4. It is convenient to first characterise average quantities. The average work done along ($A$) and ($C$) is given by the integrated power

$$\langle W_X \rangle = - \int_0^{\tau_X} dt \; \text{Tr}\left( \dot{H}[\vec{E}^{(X)}(t)] \, \rho^{(X)}(t) \right), \quad X = \{A, C\}, \tag{59}$$

where $\rho^{(X)}(t)$ is the respective solution to (53) for the two protocols $\gamma_A$ and $\gamma_C$. Similar to taking a large step approximation that was made in (18), we assume that the duration of Step ($A$) and ($C$) is large relative to the characteristic timescale $\tau^{eq}$ of the dynamics (53). In this case, by neglecting terms of second order in $\tau^{eq}/\tau$ one can approximate the corresponding entropy production (26) as

$$\langle \sigma \rangle = \beta \left( W_{\text{max}} - \langle W_A \rangle - \langle W_C \rangle \right),$$
$$\simeq \beta^2 \int_0^{\tau_A} dt \left[ \frac{d\vec{E}^{(A)}}{dt} \right]^T \mathbf{G}^{(A)}(t) \left[ \frac{d\vec{E}^{(A)}}{dt} \right] + \beta^2 \int_0^{\tau_C} dt \left[ \frac{d\vec{E}^{(C)}}{dt} \right]^T \mathbf{G}^{(C)}(t) \left[ \frac{d\vec{E}^{(C)}}{dt} \right], \tag{60}$$

where $\mathbf{G}^{(X)}$ is a symmetric $d \times d$ positive matrix given by

$$\mathbf{G}^{(X)}(t) = k_B T \; \mathbf{T}^{eq}[\vec{E}^{(X)}(t)] \circ \mathbf{F}[\vec{E}^{(X)}(t)], \tag{61}$$

where $\mathbf{F}[\vec{E}^{(X)}(t)]$ is the thermodynamic metric tensor with elements given by the negative Hessian of the free energy [28]:

$$\left( \mathbf{F}[\vec{E}] \right)_{nm} := - \frac{\partial^2}{\partial E_n \partial E_m} F(\vec{E}). \tag{62}$$

The matrix $\mathbf{T}^{eq}[\vec{E}(t)]$ is the integral relaxation tensor for the dynamical generator (53), which describes the various timescales over which the conjugate forces associated with the Hamiltonian decay to their equilibrium values. For brevity we do not specify its exact form here, though details can be found in [25, 40].

We now move to the full probability distribution $P(W)$. Assuming that the system equilibrates at the four boundary points ($\vec{E}_i$, $\vec{E}_a$, $\vec{E}_b$, and $\vec{E}_f$), we can again treat the total work extracted as a sum of three independent random variables $W = W_A + W_B + W_C$. For the approximate isothermal processes, Steps ($A$) and ($C$), the work distribution can be found through unravelling the master equation (53) in terms of the incremental changes in the energy as the system interacts with the environment (see eg. [41–43]). Since we assume that the system evolves slowly with respect to the characteristic timescales of the bath, the work distributions along Steps ($A$) and ($C$) will be of the Gaussian form (21), again with the fluctuations proportional to the dissipated work according to the fluctuation-dissipation relation (22) [30,33,44]. Therefore, in this more general setting, the probability of extracting work $W \geq \Lambda > -\Delta F$ will also be given by (27), with $\langle \sigma \rangle$ given by the more involved expression (60). Crucially, this also means that the overall principle of minimum entropy production applies in this more general setting. To minimise the entropy production we introduce the line element along Steps ($A$) and ($C$):

$$ds_X := \beta \sqrt{[d\vec{E}^{(A)}]^T \, \mathbf{G}^{(X)} \, [d\vec{E}^{(A)}]}, \tag{63}$$

with $X = \{A, C\}$ and the corresponding geodesic length

$$\mathcal{L}_X := \min_{\gamma_X} \int_{\gamma_X} ds_X . \tag{64}$$

It then follows again from the Cauchy-Schwarz inequality that the average entropy production (60) is tightly bounded according to

$$\langle \sigma \rangle \geq \langle \sigma^* \rangle := \frac{1}{\tau}\big(\mathcal{L}_A + \mathcal{L}_C\big)^2, \qquad \tau = \tau_A + \tau_C, \tag{65}$$

where we set the duration of each step to be

$$\tau_A = \tau\left(\frac{\mathcal{L}_A}{\mathcal{L}_A + \mathcal{L}_C}\right), \tag{66}$$

$$\tau_C = \tau\left(\frac{\mathcal{L}_C}{\mathcal{L}_A + \mathcal{L}_C}\right). \tag{67}$$

We therefore have the upper bound on the probability of extracting work above the free energy as

$$P(W \geq \Lambda) \leq \frac{1}{2} p(\vec{E}_a) \, \mathrm{erfc}\left(\frac{\beta(\Lambda - W_{\max}) + \langle \sigma^* \rangle}{2\sqrt{\langle \sigma^* \rangle}}\right), \tag{68}$$

where $p(\vec{E}_a)$ is the probability of having no excitation at Step B. By similar reasoning to (51), if we optimise over $\{\vec{E}_a, \vec{E}_b\}$ this bound will give a correction to the Jarzynski bound (3) that converges slower than $\sqrt{\tau^{eq}/\tau}$, where $\tau^{eq}$ is the characteristic timescale of (53). Note that, while the above bound is tight and depends only on the four boundary points, finding the exact protocol to saturate along with an analytic expression for the thermodynamic length (64) requires solving the relevant geodesic equation [25]. This will consist of a set of $d$ coupled second-order differential equations that depend on the particular structure of the generator (53). Nonetheless, our main conclusion still stands: minimising the entropy production along the slow isotherms maximises the probability of extracting work above the free energy.

# 6 Conclusions

Taking as a starting point the optimal protocols in the infinite-time limit [15], we have developed optimal finite-time protocols for maximising probabilistic violations of the second law for driven systems in contact with a Markovian thermal environment. We have obtained a general expression (68) for the corrections in terms of the so-called thermodynamic length [27, 28], and we have explicitly computed these corrections for the particular case of a driven qubit system with a simple thermalisation model (see in particular in Fig. 3). Two general insights can be obtained from our results. The first one is that protocols that minimise average entropy production can be utilised to maximise $P(W \geq \Lambda)$ by combining geodesic paths with an intermediate quench. This has enabled us to demonstrate a connection between thermodynamic length [23, 25–28] and the higher order statistics of extracted work. The second insight is that the convergence to the asymptotic bound (3) is always slower than $1/\sqrt{\tau}$, where $\tau$ is the total time of the process, which shows that finite-time effects noticeably constrain a system's ability to maximise its chance of violating the second law while respecting the fluctuation theorem (2). This should be contrasted with the faster convergence exhibited by the average dissipation which is known to scale as $1/\tau$ at leading order [45], and reflects the fact that the cumulative work distribution depends non-linearly on its average.

While the bounds and optimal protocols that we have derived are relevant for small scale systems where fluctuations are significant, they become irrelevant at larger scales due to the exponential suppression caused by the fluctuation theorem (2). However, there is a possibility to avoid the exponential decay of the likelihood $P(W \geq \Lambda > -\Delta F)$ at macroscopic levels through the use of catalysis [46]. It would interesting to apply our methods to investigate the optimal bounds on such an approach, as this could offer the possibility of increasing the probabilistic work yield from larger systems with an improved scaling with respect to the size of the working substance.

# Acknowledgments

H. J. D. M. acknowledges support from the Royal Commission for the Exhibition of 1851. M. P.-L. acknowledges funding from Swiss National Science Foundation through an Ambizione grant PZ00P2-186067.

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
