# Peer review of "Finite-time bounds on the probabilistic violation of the second law of thermodynamics"

_SciPost Physics, doi:SciPost Phys. 14, 072 (2023)_

## Round 1 · Referee Report · Anonymous (Referee 1) · 2022-8-3

Strengths

  1. Timely and accessible exposition
  2. Rigorous derivation

Report

The authors examine a scheme to maximise the probabilisitic violation of the 2nd law for finite time protocols. In particular, they start from the protocol described in Ref. 15 by Cavina, Mari and Giovannetti, who demonstrated that a particular protocol allows to realise the probability distribution that maximises the, otherwise exponentially suppressed, chances of extracting work above the free energy. The probability distribution itself takes a very special form, comprising of two delta-peaks one above and one sufficiently far below the mean work. The present contribution starts from this protocol and extends it to more realistic finite time protocols, highlighting the practical relevance of this based on recent experimental implementations. The main results demonstrate that there is a trade off between the dissipation and the ability to maximise the probabilistic violation of the second law. That such a trade off exists is perhaps intuitive, however, the present work makes this mathematically concrete for slowly driven processes by considering a piecewise dynamics of N-steps characterised by small quenches followed by rethermalisation. The authors further show that the finite time protocol can be optimised by finding the geodesic path. While the core of the manuscript deals with a single two-level system, they extend their results to arbitrary finite dimensional systems.

The work is extremely well written and in this referee's opinion is an excellent addition to the literature. While technical, the authors have presented their work in quite a pedagogical manner. I expect the present work to have impact both at the fundamental level in understanding the stochastic nature of the 2nd law and in more practical avenues in terms of understanding the optimal dynamics for a given process.

I believe the present work is suitable for publication in its current form. Below I have some comments/curiosities that I invite the authors to consider, however stress that they are optional:

  1. The double peaked distribution is reminiscent of the distribution required to saturate the thermodynamic uncertainty relation derived in PRL 123, 090604 (2019). Is this simply coincidental?

  2. When dealing with a system with multiple energy levels in Sec. V it is not clear whether the Hamiltonian in eq. 50 has degeneracies or not. Would the presence of degeneracy have any effect?

Requested changes

n/a

  • validity: high
  • significance: good
  • originality: good
  • clarity: top
  • formatting: excellent
  • grammar: perfect

Author:  Harry Miller  on 2022-11-29  [id 3087]

(in reply to Report 1 on 2022-08-03)

We thank the referee for their careful reading of the manuscript and positive comments. Regarding the the two questions raised by the referee, we provide some brief responses:

  1. The fact that the twin peaked distribution seems to connect to the saturation of the thermodynamic uncertainty relation is very interesting and we were not previously aware of this. At present we do not have an understanding of why these are related, and this certainly warrants future investigation.

  2. In Section V we have assumed no degeneracies for simplicity, and we now add a comment in the manuscript to make this concrete. If one were to allow for degenerate energy levels in the Hamiltonian, this would not effect the protocol as long as one can still guarantee that the system dynamics always has an instantaneous thermal fixed point as in Eq. 52.

---

## Round 1 · Referee Report · Anonymous (Referee 2) · 2022-11-23

Strengths

1 - An important result that quantifies ``the price" one has to pay for performing a protocol in finite time, namely what it the convergence rate of a protocol towards the Jarzynski equality.
2 - The main conclusion, which is the scaling of the convergence, is of special interest as it is different from the recently discovered universal \tau^{-1} scaling for the optimal protocol that transforms between two probabilities in the Fokker-Planck equation (Muka Nakazato and Sosuke Ito from 2021).
3 - The manuscript is well written and easy to follow.

Weaknesses

The derivation is probably not the most mathematically elegent way to show the results.

Report

In this manuscript, the authors considered the following questions: What are the fundamental limitations on the probabilistic violation of the Jarzynski relations in finite time? What are the corresponding optimal protocols? How are they related to optimal protocols for maximising the average work?

These questions are solved for system with a discrete set of energies, that is in contact with a Markovian bath. This is done by considering a general protocol as a sequence of quenches followed by relaxations, and calculating the answer in such protocols in the limit of a large number of such quenches. This is correct, but my guess is that there is probably a better way to argue the same with the need to discretize the protocol, maybe using results from calculus of variation. But this is just a matter of elegance - as far as I could tell the results are correct.

It will be very interesting to compare the final scaling in this case to the results on the entropy production in arbitrary probability change for a Fokker-Planck equation, recently obtained in ``Geometrical aspects of entropy production in stochastic thermodynamics based on Wasserstein distance", by Muka Nakazato and Sosuke Ito. The Jarzynski equation should hold also for the Fokker-Planck scenario, and although the object of optimization is not identical, the different scaling is nevertheless surprising. It will be very interesting to compare the two results, and understand where the different scaling come from. Is this a consequence of the discreteness assumed in the current manuscript, or maybe from the different objective for the optimization?

Requested changes

I could find a single typo that should be fixed in the manuscript -- look for `beyond the beyond the' in the introduction.

  • validity: high
  • significance: good
  • originality: good
  • clarity: good
  • formatting: excellent
  • grammar: excellent

Author:  Harry Miller  on 2022-11-29  [id 3088]

(in reply to Report 2 on 2022-11-23)

We are thankful for the referee's report and supportive comments. Here we would like to respond to some of the points and suggestions raised.

Firstly, we would like to highlight that it is not necessary to discretize the protocol to arrive at our general results. This is shown in Section V, where we use a general continuous time master equation to rederive the results obtained through the discrete approach. This would also encompass a classical Focker-Planck dynamics as well. The key necessary steps are to assume a dynamics with a thermal fixed point, and ensure the system is close-to-equilibrium at all times in Step A and C.

Secondly, the referee correctly points out that the average dissipation scales differently to the cumulative distribution. We would like to stress that this is not a result of discretising the protocol, and instead comes from the fact that the cumulative distribution depends non-linearly on the average dissipation, as shown eg. in Eq. 25. We have added a brief comment on this point in the final discussion as well as a reference to the paper of Nakazato and Ito.

---

## Round 2 · Author Response

We thank the referees for their careful reading of the manuscript and constructive comments. We have now added some additional clarifications in the paper regarding some of the referee remarks as listed below.

---

## Round 2 · List of Changes

1. We have specified that we assume no degeneracies in the system Hamiltonian, below Eq. (50).

  2. A remark about the different scalings for average dissipation and the cumulative distribution has been added to discussion, alongside an additional reference [46].

  3. Corrected a minor typo in Eq. (66).

---

## Editorial Decision

published